# Temporal Change in Biomarkers of Bone Turnover Following Late Evening Ingestion of a Calcium-Fortified, Milk-Based Protein Matrix in Postmenopausal Women with Osteopenia

**DOI:** 10.3390/nu11061413

**Published:** 2019-06-23

**Authors:** Manjula Hettiarachchi, Rachel Cooke, Catherine Norton, Phil Jakeman

**Affiliations:** Health Research Institute, University of Limerick, V94 T9PX Limerick, Ireland; manjula.hettiarachchi@ul.ie (M.H.); rachel.cooke@gmail.com (R.C.); catherine.norton@ul.ie (C.N.)

**Keywords:** randomized control trial, milk protein matrix, nutrient intervention, nutrient timing, bone remodeling, bone turnover markers, bone health, osteopenia, postmenopausal women

## Abstract

The diurnal rhythm of bone remodeling suggests nocturnal dietary intervention to be most effective. This study investigated the effect of bedtime ingestion of a calcium-fortified, milk-derived protein matrix (MBPM) or maltodextrin (CON) on acute (0–4 h) blood and 24-h urinary change in biomarkers of bone remodeling in postmenopausal women with osteopenia. In CON, participants received 804 ± 52 mg calcium, 8.2 ± 3.2 µg vitamin D and 1.3 ± 0.2 g/kg BM protein per day. MBPM increased calcium intake to 1679 ± 196 mg, vitamin D to 9.2 ± 3.1 µg and protein to 1.6 ± 0.2 g/kg BM. Serum C-terminal cross-linked telopeptide of type I collagen (CTX) and procollagen type 1 amino-terminal propeptide (P1NP), and urinary N-telopeptide cross-links of type I collagen (NTX), pyridinoline (PYD) and deoxypyridinoline (DPD) was measured. Analyzed by AUC and compared to CON, a −32% lower CTX (*p* = 0.011, d = 0.83) and 24% (*p* = 0.52, d = 0.2) increase in P1NP was observed for MBPM. Mean total 24 h NTX excreted in MBPM was −10% (*p* = 0.035) lower than CON. Urinary PYD and DPD were unaffected by treatment. This study demonstrates the acute effects of bedtime ingestion of a calcium-fortified, milk-based protein matrix on bone remodeling.

## 1. Introduction

Low bone mineral density (bone mineral content) and a diminution in bone quality (bone microarchitecture) are attributes of risk of fracture in people with osteopenia [1]. Bone architecture and mineral content remodel continuously throughout life with high rate of bone tissue remodeling an independent factor of fracture risk [2]. In the treatment of people with osteopenia, measurement of change in remodeling rate through intervention informs clinical practice [3]. Diet is a modifiable, lifestyle factor that can affect bone health and the risk of fracture. Therefore, a nutritional intervention that positively modulates the linked remodeling processes of bone resorption and formation is an attractive option in the maintenance of bone health.

Bone turnover markers (BTM) of bone resorption (C-terminal cross-linked telopeptide of type I collagen, CTX) and formation (total procollagen type 1 amino-terminal propeptide, P1NP) inform of change in bone metabolism and bone health earlier, and with greater sensitivity than change in bone mineral density (BMD) [4]. Defined by circadian variation in BTM, bone remodeling exhibits a unimodal diurnal rhythm with a nocturnal peak and daytime nadir [5,6,7,8]. The diurnal amplitude of bone resorption is greater than of the bone formation and is, thereby, considered the more prominent and sensitive biomarker of change in bone remodeling. The ratio of biomarkers of bone formation/resorption (i.e., P1NP/CTX ratio) tracks the temporal balance of bone remodeling throughout the day [5].

Bone remodeling is sensitive to modulation by feeding and nutrient specific intake. The response of bone remodeling to food ingestion is linked putatively to the secretion of enterogastric hormones glucose-dependent insulinotropic peptide (GIP_1–42_) and glucagon-like peptide-1 (GLP-1_7–36_) acting via an entero-osseous axis [9]. The antiresorptive effect of a postprandial rise in GIP and GLP-1 acts to decrease osteoclast activity [10] resulting in a ~20% decrease in CTX in healthy adults [11,12]. Specific nutrient intake includes calcium and vitamin D, key nutrients for bone health [13]. Intake of calcium and vitamin D suppress parathyroid activity leading to decreased parathyroid hormone (PTH) secretion and reduction in bone resorption activity [14]. Dietary and/or supplemental intake of these nutrients is advised by reference to population-based recommended intake *per diem.* Few nutritional studies of bone metabolism time supplemental intervention to act at the nocturnal peak of bone remodeling [15].

The matrix of nutrients within dairy-based food products are potent comodulators of bone remodeling [16]. Superior to an individual nutrient, a food matrix contains multiple nutrients of individual biofunctionality that are pluripotent in effect when consumed as a whole [17]. Timed to act during the nocturnal peak rate of bone remodeling, we report the outcome of a randomized control trial investigating the effect of feeding a milk protein-based matrix (MBPM) fortified with calcium and vitamin D prior to bedtime on the acute (0–4 h) and 24 h change in biomarkers of bone remodeling in postmenopausal women with osteopenia. An isoenergetic maltodextrin acted as a control (CON).

## 2. Materials and Methods

The study was approved by the University of Limerick, Faculty of Education and Health Sciences Research Ethics Committee (2017_06_03_EHS). Participants were informed of the purpose of the study and all known risks before providing written, informed consent. The study was conducted in accordance with the ethical standards outlined in the most recent version of the Declaration of Helsinki and registered with clinicaltrials.gov identifier NCT03337971.

### 2.1. Participant Recruitment and Progression

A convenience sample of 41 postmenopausal women aged 50 to 70 y recruited from the University of Limerick Body Composition (ULBC) Study database by invitation. ULBC subjects had preassigned alphanumeric ID codes that continued to the present study. Subject volunteers were free-living, lactose tolerant, and devoid of treatment for osteoporosis or drugs affecting calcium absorption. Dual energy X-ray absorptiometric (DXA; Lunar iDXA™ with enCORE™ 2007 v.17 software, GE Healthcare, Chalfont St Giles, Bucks., UK) measurement of AP spine (L1–L4) and dual femur BMD (femoral neck), conducted and analyzed in accordance with the International Society for Clinical Densitometry [18], confirmed eligibility (BMD T scores < −1.0, ≥ −2.5). Twenty-four volunteers confirmed as osteopenic submitted to a clinical examination, blood screen and 7-day weighed food dietary intake record. Following screening five volunteers declined to participate further and two excluded based on medication (treatment for osteoporosis: *n* = 1; medication with Gabapentin™: *n* = 1). One participant was unable to attend during the intervention periodThe remaining 16 volunteers progressed to the RCT and randomly assigned to receive either a milk-based protein supplement (MBPM) or an isoenergetic, maltodextrin control (CON) over two consecutive days (Figure 1). Appendix A provides a CONSORT flowchart and checklist for this randomized trial.

### 2.2. Study Design

A block-randomized, within-subject crossover design examining acute (0–4 h) and 24 h change in biomarkers of bone turnover in healthy, postmenopausal women with osteopenia following ingestion of MBPM or CON was conducted. Estimated from a least significant change in bone resorption (i.e., a 30% reduction in CTX with feeding) statistical power (1-β) of 80% and two-sided α of 0.05 would be achieved by a sample size of 16 participants.

### 2.3. Dietary Intake and Standardized Diets for the Intervention

Following instruction from a qualified dietitian, participants recorded their dietary intake for seven consecutive days (5 weekdays, 2 weekend days) using a weighed food and fluid log. Participants reported food type, brand name, quantity, cooking method, type, time, and location of meal preparation. Subsequent coding and analysis was completed using Nutritics^™^ Dietary Analysis Software (Nutritics^©^ software version 4.0 for Ireland). Dietary intake was compared to reference intakes (DRIs) developed by the Food and Nutrition Board (FNB) at the Institute of Medicine (IOM) of The National Academies [19].

Based on the 7-day dietary intake data a 48 h dietary standardization, representative of the individual’s reported habitual dietary patterns of the research population, compliant with dietary recommendations (IOM DRIs) for energy, protein, and calcium, reflective of participants’ food preferences and normalized to body mass was prescribed for each participant by a registered dietitian. Each participant received the same meal plan on each of day of the intervention. Individuals’ standardized diet was supplemented with CON or MBPM. At bedtime, participants received a supplemental intake of MBPM or CON. The nutrient composition of MBPM and CON (Appendix A) comprised an additional 0.3 g/kg body mass protein and 0.3 g/kg body mass carbohydrate in the MBPM and an equivalent amount of carbohydrate energy, i.e., 0.6 g/kg body mass the CON arm of the study. Appendix A report the prescribed dietary intakes inclusive of either MBPM or CON.

### 2.4. Trial Protocol and Procedures

Participants were resident at the University of Limerick during the consecutive 2-day, 2-night trial (Figure 2). Participants were free-living but remained with the confines of the university campus and refrained from moderate to strenuous physical activity during their stay. Participants arrived at 17:00 h and settled into the residence before emptying the bladder. This marked the 24 h urine collection starting point for day 1 and the end sampling point 24 h later. Each participant received standardized meals and the same meal options at the same time points over both days. At 20:00 h a research nurse inserted a cannula into a superficial arm vein and a blood draw (5 mL) was taken. Each participant consumed either MBPM or CON at 22:00 h and retired to bed. Serial blood draws (5 mL) each hour for a further 4 h and the cannula removed.

Serum and plasma were separated by centrifugation at 10,000× *g* at 4 °C for 5 min and frozen at −80 °C until analysis. Sixteen of the 192 blood samples drawn were hemolyzed and not analyzed. This reduced the full analysis from 3 participants. Active plasma GLP-1(7-36) and total GIP (1-42) was measured using the MSD^®^ metabolic assay kits (Meso Scale Discovery, Rockville, MD, USA) based on sandwich ELISA and according to the manufacturer’s instructions. The inter-assay CVs were 7.3 and 16.1 for active GLP-1 and total GIP, respectively. Plasma CTX and serum P1NP by 2-site immunometric assay using electrochemiluminescent detection (Roche Cobas e411, Roche Diagnostics, UK). Plasma parathyroid hormone (PTH) measured using the human bone magnetic bead panel-1 kit (cat3 HBNMAG-51k, Millipore Corporation, Billerica, MA, USA) employing Luminex^™^ technology. The interassay CV was 5.3% for CTX, 4.5% for P1NP and <10% for PTH. 24 h urine collection commenced at 17:00 h on day 1, following full void, to 17:00 h of the 2nd day. An aliquot of urine was frozen at −20 °C until batch analysis for urinary free pyridinoline (fPYD) and deoxypyridinoline (fDPD) by LC-MS/MS [20], urinary N-terminal telopeptide of Type I collagen (uNTX) by commercial immunoassay and creatinine (Jaffé method, Roche COBAS^®^ C501, Roche, Burgess Hill, UK). Urinary assays were undertaken by the Bioanalytical Facility, University of East Anglia, Norwich, UK.

### 2.5. Statistical Analysis

Shapiro–Wilk test was applied to check for normality of the data that are presented as the mean ± SD unless stated otherwise. Temporal change in serially sampled data was analyzed by repeated measures ANOVA, the level of significance for post-hoc tests subject to Bonferroni correction. Difference between mean change attributed to CON and MBPM was analyzed by paired *t*-test. Cohen’s effect size was calculated using the standard formula (*t*-score/√n). Statistical significance was set at *p* ≤ 0.05.

## 3. Results

### 3.1. Subject Characteristics

The baseline characteristics of the subject population are representative of postmenopausal women age 55 to 70 y (Table 1). Subjects confirmed as osteopenic based on the BMD T-score of the L1-L4 region of the spine.

### 3.2. Dietary Intake

Reported mean daily intake, from a 7-day food and fluid log (Appendix A) demonstrated adequacy of reported intakes among 81% of the study population for dietary protein (1.13 ± 0.4 g/kg/d) when assessed relative to IOM RDA (0.8 g/kg/d). Forty-four percent reported mean daily intakes of protein above 1.2 g/kg/d. The mean reported daily calcium intake (873 ± 366 mg) surpassed the lowest acceptable intake (800 mg) for 50% of participants with only 25% of the population reporting calcium intakes at or above the IOM RDA of 1200 mg. None of the study population met the current IOM RDA for vitamin D (15 µg). Nine participants (56%) reported a mean daily vitamin D intake below 5 µg per day.

Appendix A provides individuals’ standardized nutrient intake for MBPM and CON. MBPM and CON augmented individuals’ standardized total energy intake by 9% (to a mean energy intake of approximately 2000 kcal). In addition, MBPM increased protein intake by 25% (to 1.63 ± 0.19 g/kg/d), calcium by 109% (from 804 ± 153 to 1679 ± 196 mg) and vitamin D 12% (from 8.2 ± 3.2 to 9.2 ± 3.1 µg) (Table 2).

### 3.3. Enterogastric Response to Feeding

Feeding induced a differential response for GIP compared to GLP-1. The temporal change in GIP was to increase by 40% from 109 ± 32 pmol/L to a peak of 152 ± 37 pmol/L 60 min postingestion in CON and by 50% from 99 ± 30 pmol/L to a peak of 150 ± 39 pmol/L 60 min postingestion in MBPM. Thereafter, GIP decreased to 60 ± 17 pmol/L residing 45% lower than prefeeding 4 h postingestion. Analyzed by AUC, no difference in overall response between CON and MBPM was observed (Figure 3a).

In contrast, GLP-1 decreased in a linear manner with time postfeeding in both trials. After 4h GLP-1 decreased by 38% from 65 ± 16 pmol/L prefeeding to 40 ± 11 pmol/L after 4 h postingestion in CON and by 30% from 72 ± 16 pmol/L to 51 ± 12 pmol/L 4 h postingestion in MBPM. No difference in overall response between CON and MBPB was observed (Figure 3b).

### 3.4. Parathyroid Hormone and Bone Turnover Marker Response to Feeding

Feeding induced a differential PTH response for CON compared to MBPM. PTH increased by 15% from 3.53 ± 1.42 ng/mL to a peak of 4.0 ± 1.48 ng/mL 4 h postingestion in CON and decreased by 14% from 3.37 ± 1.44 ng/mL to a nadir of 2.9 ± 1.28 ng/mL 3 h postingestion in the MBPM arm of the trial. Analyzed by AUC, a 2.3-fold lower response to MBPB than CON was observed (mean difference −109 ng·min/mL; 95% CI −170, −48; *p* = 0.002, d = 1.08) (Figure 4a).

Serum BTMs, CTX, and P1NP increased following feeding. The temporal change in CTX increased by 43% from 0.32 ± 0.18 ng/mL to a peak of 0.45 ± 0.23 ng/mL 4 h postingestion in CON and by 21% from 0.33 ± 18 ng/mL to peak at 0.43 ± 0.25 ng/mL 4 h postingestion in the MBPM arm of the trial. Analyzed by AUC, a −32% lower CTX response to MBPB than CON was observed (mean difference −3.39 ± 4.0 ng·min/mL; 95%CI −5.84, 0.93; *p* = 0.011, d =0.83) (Figure 4b).

The temporal change in P1NP was a mean increase of 16% from 44.4 ± 19 ng/mL to peak at 51.3 ± 21 ng/mL 4 h postingestion in CON and by 19% from 40.8 ± 19 ng/mL to a peak of 48.3 ± 21 ng/mL 4 h postingestion in the MBPM arm of the trial. Analyzed by AUC, a 24% greater response to MBPB than CON was observed (mean difference 116 ± 599 ng·min/mL; 95%CI −265, 497; *p* = 0.52, d = 0.2) (Figure 4c).

### 3.5. Urinary 24 h BTM Excretion

Participants’ urine output varied from −750 to 1100 mL, resulting in a 4.1 ± 23% mean difference in total 24-h urine volumes between days. However, total 24-h urinary creatinine excretion was not significantly different between days (mean difference 0.0063 ± 0.17 g; 95%CI −0.085, 0.097; *p* = 0.89), providing confidence that total urine collection was attained.

A large interindividual variation in 24 h urinary BTM excretion was observed (Figure 5). Mean total 24 h NTX excreted for MBPM was −10 ± 16% lower than for CON (mean difference −43.2 ± 74.5 nmol BCE; 95%CI −82.9, 3.5, *p* = 0.035, d = 0.58), Figure 5a. No significant difference was observed in mean 24 h excretion of PYD (219 ± 98 vs. 256 ± 107 nmol) or DPD (43.0 ± 24.6 vs. 47.3 ± 27.4 nmol) for CON (Figure 5b) and MBPM (Figure 5c), respectively.

## 4. Discussion

The reported baseline dietary intakes of the present cohort of Irish women with osteopenia were comparable to results from the Irish National Adult Nutrition Survey (NANS) [21]. The mean total energy (kcal) consumed per day by our population differed by only 6% to the mean energy intake in the NANS study (1552 ± 382 kcal). The macronutrient contribution to total energy varied slightly in our study (carbohydrate 40.4 ± 9.2%, protein 19.4 ± 6.4% and fat 36.3 ± 9.9%) vs. the NANS study (carbohydrate 46 ± 5.9%, protein 18.3 ± 2.8%, and fat 35.5 ± 6.9%). The mean calcium content of the baseline dietary record (876 ± 443 mg/day) exceeded the AR for calcium. However, these data are distorted by the intake of calcium-containing dietary supplements, with only 56% (*n* = 9) of participants actually achieving this by food intake alone. A difference of 11% was observed in dietary calcium intake between our study sample and NANS data (MDI = 995 ± 573 mg/day) [10]. The habitual dietary intake data confirmed adequacy of energy and essential nutrients and allowed an individual standardized meal plan to be designed reflecting habitual intake over the 2-day study. The main effect of the bed-time supplement was to increase mean daily, and specifically timed bed-time, energy, protein, and calcium intake, by 9, 25, and 109%, respectively, in MBPM, and a matched energy intake of 9% in CON. A nonhabitual bedtime supplemental mean intake of energy (166 kcal), protein (22.2 g), and calcium (875 mg) was the nutrient intervention ascribed to MBPM.

The principal outcome of this block randomized, within subject crossover trial in postmenopausal women with osteopenia was to effect a positive change in bone remodeling by nocturnal feeding. Timed to coincide with peak rate of bone remodeling MBPM effected an acute (4 h) reduction in the serum bone resorption marker CTX, a corresponding increase in the bone formation marker P1NP and a moderately greater reduction in the P1NP/CTX ratio (mean difference −3.46; 95% CI −7.99, 1.07, *p* = 0.12, d = 0.46) compared to CON. Though moderate in magnitude, the outcome of the nocturnal feed was a transient reduction in the nocturnal peak of bone remodeling.

Ingestion of food increases enterogastric secretion of GIP and GLP-1 that are putative regulators of bone metabolism acting on GIP and GLP-1 and 2 receptors located on bone resorbing osteoclasts and bone forming osteoblasts [10,11,12,13,14]. CON and MPBP induced a similar enterogastric response. GIP increased earlier, and to greater extent, than GLP-1 because GIP is secreted primarily from the proximal duodenum, whereas GLP-1 is secreted from cells located primarily in the distal jejunum and ileum. The lack of a differential response to secretion of incretins by MBPM over CON is surprising. Milk-based protein increases the secretion of incretin hormones [22] and preserves incretin bioactivity through the inhibition of dipeptidyl peptidase (DPP-IV) activity [23,24]. Furthermore, having similar calcium content to MBPM (~850 mg) for co-ingestion with a mixed macronutrient meal has been shown to augment the postprandial incretin response [25], in particular GIP (+47%), probably due to the greater increase in calcium concentration in the duodenum than the ileum in the immediate period postingestion.

The age-related increase in serum PTH levels with age is one attribute of age-related increase in bone resorption. The secretion of PTH, a principal regulator of the quantity of remodeling activity, responds to change in serum ionized calcium and is very much evident in the differential change in circulating PTH between CON and MBPM. It follows that calcium intake induces a remodeling transient and markers of bone metabolism react accordingly [26]. Though not timed specifically to the nocturnal peak, provision of an oral calcium load of 0.2 g to late postmenopausal women with normal calcium absorption significantly decreased serum CTX, for up to 5 h postingestion and can be prolonged further by increasing the calcium load. [27]. Earlier studies designed to eliminate nocturnal PTH transients by 24 h calcium infusion, or by anti-PTH antibodies, suggest that PTH may not mediate the circadian pattern of bone resorption, but potentially influence the absolute rate of bone resorption at which this pattern occurs [28]. Nutrition-induced nocturnal reduction in PTH would likely be beneficial to an adaptive (reduced) rate of bone remodeling and preservation of bone health.

Exhibiting higher diurnal amplitude, the overall influence on circadian, 24-h bone resorption is informed by the measurement of 24 h urinary N-telopeptide cross-links of type I collagen (NTX), and associated cross-links pyridinoline (PYD) and deoxypyridinoline (DPD). A dilution of the acute (0–4 h), transient change in BTM response to nocturnal feeding within a 24 h cycle was expected, but only 24 h urinary NTX reflected the circulatory changes in BTM in the acute phase postfeeding. Furthermore, the associated response of CTX (AUC_0-240_) and 24 h urinary NTX were significantly correlated (r = 0.71, *p* < 0.001), whereas DPD (r = 0.022, *p* = 0.91) and PYD (r = 0.012, *p* = 0.9) were not. Earlier studies demonstrate the value of urinary cross-links to monitor the circadian rhythm of bone resorption [6,29], so the lack of treatment effect on the bone specific pyridinoline cross-links was surprising. Indeed, we observed an almost 10-fold, intersubject 24 h excretion and high variability and sensitivity of response to nocturnal feeding (Figure 5).

Protein constitutes ~50% of the volume of bone and approximately one-third of its mass. Therefore, the turnover rate of Type I collagen is important to the maintenance of bone structure and bone health. Similar to skeletal muscle, collagen turnover is a sensitive to nutritional modulation. Stable isotope studies of collagen synthesis report an increase in protein synthetic rate by ~66% following intravenous nutrition (glucose, lipid emulsion, and amino acids in the ratio of 55%:30%:15% energy, respectively) [30]. Controversy exists regarding recommendations for amount and type of dietary protein for bone health [31]. Evidence in support of dairy protein intake is provided in a recent review [32]. Equally, there is a strong link between calcium supplementation, particularly dairy-based calcium, and bone health, especially in the elderly linked to the high bioavailability of calcium and encrypted bioactive peptides/amino acids or proteins with the potential to improve calcium absorption and regulate bone metabolism [33].

## 5. Conclusions

In summary, the reduction in AUC of CTX (−30% compared to CON) confirms our initial hypothesis that a dairy-based protein supplement fortified with calcium (MBPM) fed at bedtime has a potent effect on nocturnal rates of bone resorption in healthy osteopenic postmenopausal women. The synergistic, pluripotent quality of a milk-based protein matrix and timing of ingestion to the nocturnal, peak rate of bone remodeling transiently depressed bone turnover. We conclude that a late-evening supplement of calcium-fortified milk protein affects a beneficial decrease in the homeostatic rate of bone remodeling in persons at risk of degenerative bone disease. A 24-w longitudinal study (clinicaltrials.gov identifier NCT03701113) is currently in progress to assess the efficacy of outcome in bone turnover and BMD in this population.

## Figures and Tables

**Figure 1 nutrients-11-01413-f001:**
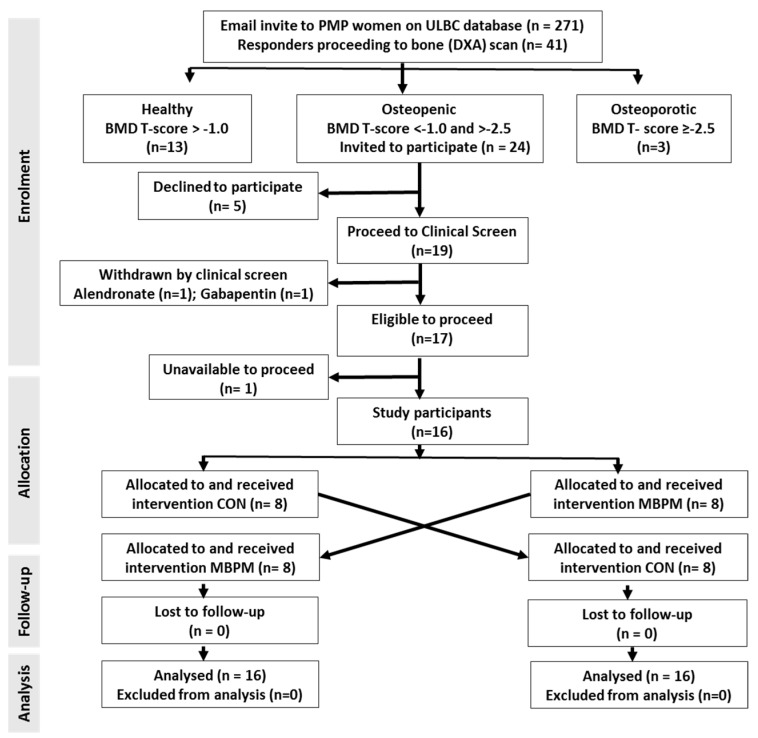
Flow diagram of participant enrolment, allocation, follow-up, and analysis.

**Figure 2 nutrients-11-01413-f002:**
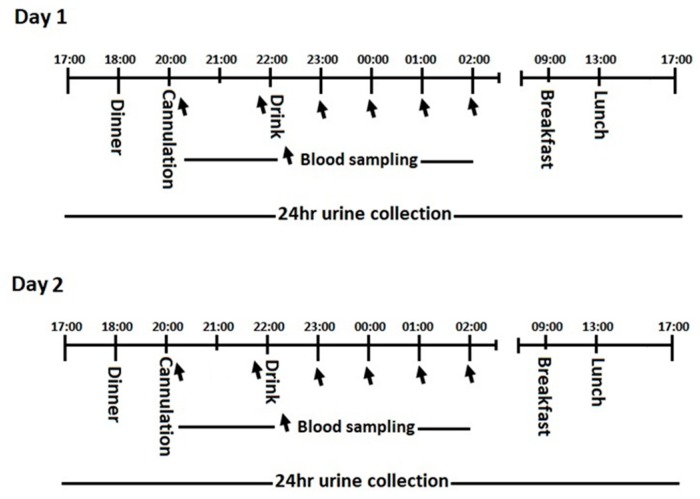
Experimental protocol.

**Figure 3 nutrients-11-01413-f003:**
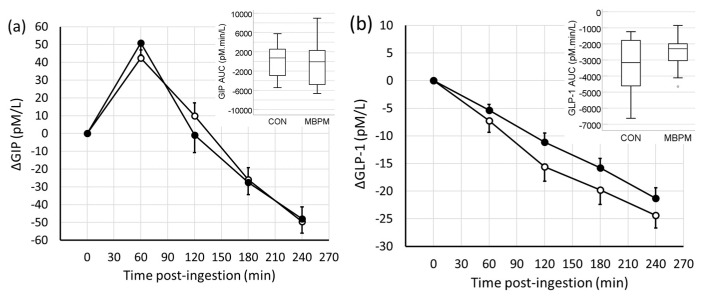
Temporal change in enterogastric peptides following ingestion of CON (open circles) and MBPM (filled circles); (**a**) GIP and (**b**) GLP-1. Data are the mean and SEM, *n* = 13. Insert denotes the area under the curve, AUC_0-240._

**Figure 4 nutrients-11-01413-f004:**
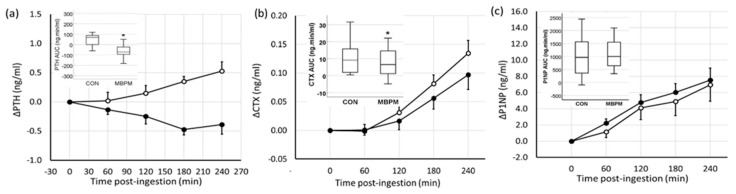
Temporal change in parathyroid hormone (PTH) and biomarkers of bone turnover following ingestion of CON (open circles) and MBPM (filled circles). (**a**) PTH, (**b**) C-terminal cross-linked telopeptide of type I collagen (CTX), and (**c**) P1NP. Data are the mean and SEM, *n* = 13. * denotes *p* < 0.05. Insert denotes the area under the curve, AUC_0-240._

**Figure 5 nutrients-11-01413-f005:**
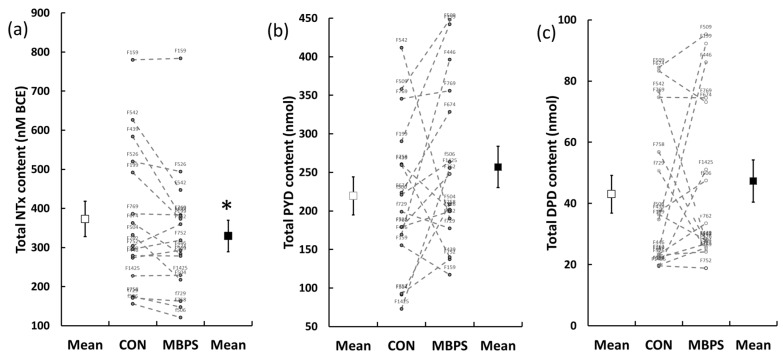
Individual and mean 24-h urinary excretion of urinary biomarker of bone turnover in the CON (open circles) and MBPM (filled circles): (**a**) total NTX, (**b**) total PYD, and (**c**) total DPD. Data are the mean and SEM, *n* = 16; * denotes *p* < 0.05. Alphanumeric numbers identify individual subjects’ response.

**Table 1 nutrients-11-01413-t001:** Subject population (*n* = 16).

Characteristic	Unit	Mean	SD	Range
Age	y	64.7	3.3	55.8 to 69.5
Height	cm	160.8	7.0	149.5 to 177.7
Weight	kg	67.2	10.8	48.0 to 91.1
Body mass index	kg·m^−2^	26.0	4.1	19.9 to 37.1
Fat mass	kg	26.4	7.1	14.3 to 40.1
Lean mass	kg	38.6	4.2	31.7 to 49.1
Body fat	%	38.8	4.8	29.9 to 48.3
Dual Femur	T-score	−0.81	0.49	−1.5 to −0.05
BMDspine (L1–L4)	T-score	−1.65	0.47	−2.4 to −1.0

**Table 2 nutrients-11-01413-t002:** Mean daily nutrient intake for the calcium-fortified, milk-derived protein matrix (MBPM) and maltodextrin (CON) trials (*n* = 16).

	**Mean Daily Intake in MBPM Trial**						
	Energy (kcal)	Protein (g)	Protein (g/kg)	Carbohydrate (g)	Fat (g)	Calcium (mg)	Vitamin D (ug)
Mean	2009	107.3	1.6	177.8	89.7	1679	9.2
SD	44	6.8	0.2	22.4	9.7	196	3.1
Min	1933	94.9	1.3	144.4	77.3	1345	3
Max	2079	120.3	2	217.2	107.7	2095	12.3
	**Mean Daily Intake in CON Trial**						
Mean	1994	85.2	1.3	203.7	89.7	804	8.2
SD	43	5.2	0.2	24	9.7	153	3.2
Min	1921	79.1	1	168.9	77.3	517	1.7
Max	2064	95.2	1.6	252.3	107.7	1013	11.2

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
