# Peer review of "Temporal Change in Biomarkers of Bone Turnover Following Late Evening Ingestion of a Calcium-Fortified, Milk-Based Protein Matrix in Postmenopausal Women with Osteopenia"

_nutrients, 2019, doi:10.3390/nu11061413_

Round 1
Reviewer 1 Report
This is a crossover study in 16 healthy postmenopausal women comparing the effect on bone turnover markers and parathyroid hormone of a single dose of calcium and vitamin D-fortified milk protein against energy matched maltodextrin. Outcome measures were assessed at 0-4 hours and 24 hours. An immediate decrease in markers of bone turnover and PTH was seen on the intervention day compared to control; urinary markers of bone turnover at 24 hours did not show a significant change. A longitudinal study with BMD outcomes is in progress and will add further information. Some comments follow.
Study design. Several studies have shown an effect both acutely and longer term on bone turnover and BMD in non calcium fortified milk protein (for example Aoe et al. Biosci Biotechnol Biochem 2001;65:913-8). It has also been shown that milk protein increases calcium absorption and urinary calcium excretion. The current design includes both an extra milk protein component and calcium and vitamin D fortification: it may be useful to see what additive effect if any the fortification has.
Abstract p1 line 13 has a typographical error in calcium content for the control group; should be 804 mg not 8041 mg from the text.
Abstract p1 line 16; "24h" I assume refers to 24h urine.
Abstract p1 line 20. Urinary NTX decrease of 24% should not be classed as significant either by p value or by literature least significant change.
The last paragraph of the introduction (p2) reiterates the study findings from the abstract. This looks like it belongs in the results section.
Were the power calculations determined befoe the study began?
Was there any relationship between the measured incretin hormones and either markers or PTH? Unclear why this data is included otherwise.
Was there a difference in effect of the test drink in women taking calcium supplements? Numbers may be too small to establish this.
Author Response
Response to Reviewer 1 Comments
Comments and Suggestions for Authors
This is a crossover study in 16 healthy postmenopausal women comparing the effect on bone turnover markers and parathyroid hormone of a single dose of calcium and vitamin D-fortified milk protein against energy matched maltodextrin. Outcome measures were assessed at 0-4 hours and 24 hours. An immediate decrease in markers of bone turnover and PTH was seen on the intervention day compared to control; 1.urinary markers of bone turnover at 24 hours did not show a significant change. A longitudinal study with BMD outcomes is in progress and will add further information. Some comments follow.
Study design. 2.Several studies have shown an effect both acutely and longer term on bone turnover and BMD in non-calcium fortified milk protein (for example Aoe et al. Biosci Biotechnol Biochem 2001;65:913-8). It has also been shown that milk protein increases calcium absorption and urinary calcium excretion. The current design includes both an extra milk protein component and calcium and vitamin D fortification: it may be useful to see what additive effect if any the fortification has.
3.Abstract p1 line 13 has a typographical error in calcium content for the control group; should be 804 mg not 8041 mg from the text.
4.Abstract p1 line 16; "24h" I assume refers to 24h urine.
5.Abstract p1 line 20. Urinary NTX decrease of 24% should not be classed as significant either by p value or by literature least significant change.
6.The last paragraph of the introduction (p2) reiterates the study findings from the abstract. This looks like it belongs in the results section.
7.Were the power calculations determined befoe the study began?
8.Was there any relationship between the measured incretin hormones and either markers or PTH? Unclear why this data is included otherwise.
9.Was there a difference in effect of the test drink in women taking calcium supplements? Numbers may be too small to establish this.
Response: Thank you for a positive review of this m/s.
Point 1: 1.urinary markers of bone turnover at 24 hours did not show a significant change.
Response 1: No change in PYD and DPD was observed, 24 h uNTX did decrease and correlated to AUC(0-240) CTX.
Point 2: 2.Several studies have shown an effect both acutely and longer term on bone turnover and BMD in non-calcium fortified milk protein (for example Aoe et al. Biosci Biotechnol Biochem 2001;65:913-8). It has also been shown that milk protein increases calcium absorption and urinary calcium excretion. The current design includes both an extra milk protein component and calcium and vitamin D fortification: it may be useful to see what additive effect if any the fortification has.
Response 2: We agree with the reviewer that evidence exists of the independent effect of milk protein, calcium and vitamin D. Indeed, the design of the MPM takes full advantage of these combined effects and optimises their application from extant knowledge of the diurnal rhythm of bone remodeling. It would be informative, but to undertake a comprehensive study of the effect of all components of the MBPM fortification, singly and in all combinations, would be a major challenge, not addressed by this study.
Rather than review all effects independently in the introduction we provide reference to relevant reviews related to dairy-based protein (e.g. refs 15, 16) rather than dietary protein per se (e.g. Rizzo et al /doi.org/10.1007/s00198-018-4534-5) and emphasise the matrix effect vs. single nutrient supplement (e.g. ref 17). The MBP trial promoted by the Japanese group is a single supplement (40mg per day) undertaken in healthy young women and we felt that, while relevant (as would many other articles), this paper was not specific to the short time course, matrix product or population used in this study. We have, however, made reference to this paper in the longitudinal study.
Point 3: 3.Abstract p1 line 13 has a typographical error in calcium content for the control group; should be 804 mg not 8041 mg from the text.
Response 3: Thank you. Corrected in the revision.
Point 4: 4.Abstract p1 line 16; "24h" I assume refers to 24h urine.
Response 4: Thank you. Corrected in the revision.
Point 5: 5.Abstract p1 line 20. Urinary NTX decrease of 24% should not be classed as significant either by p value or by literature least significant change.
Response 5: Thank you. A transcription error of NTX data reported in the results (lines 216-217). The revised abstract (below) provides clarity and consistency in the summary report of the data within the word limit. Corrected in the revision.
Abstract: The diurnal rhythm of bone remodeling suggests nocturnal dietary intervention to be most effective. This study investigated the effect of bedtime ingestion of a calcium fortified, milk-derived protein matrix (MBPM) or maltodextrin (CON) on acute (0-4 h) blood and 24 h urinary change in biomarkers of bone remodeling in post-menopausal women with osteopenia. In CON participants received 804 ± 52 mg calcium, 8.2 ± 3.2 µg vitamin D and 1.3 ± 0.2 g/kg BM protein per day. MBPM increased calcium intake to 1679 ± 196 mg, vitamin D to 9.2 ± 3.1 µg and protein to 1.6 ± 0.2 g/kg BM. Serum C-terminal crosslinked telopeptide of type I collagen (CTX) and procollagen type 1 amino-terminal propeptide (P1NP) and urinary N-telopeptide cross-links of type I collagen (NTX), pyridinoline (PYD) and deoxypyridinoline (DPD) was measured. Analysed by AUC and compared to CON, a -32% lower CTX (p = 0.011, d = .83) and 24% (p = 0.52, d = 0.2) increase in P1NP was observed for MBPM. Mean total 24 h NTX excreted in MBPM was -10% (p = 0.035) lower than CON. Urinary PYD and DPD were unaffected by treatment. This study demonstrates the acute effects of bedtime ingestion of a calcium fortified, milk based protein matrix on bone remodeling.
Point 6: 6.The last paragraph of the introduction (p2) reiterates the study findings from the abstract. This looks like it belongs in the results section.
Response 6: This is a carry-over from previous Journal style. As all the information contained within the paragraph (lines 63 to 69) is reproduced within the main body of the text this may be deleted. Corrected in the revision.
Point 7: 7.Were the power calculations determined before the study began?
Response 7: Yes, as stated within the study design (lines 97 to 99).
Point 8: 8.Was there any relationship between the measured incretin hormones and either markers or PTH? Unclear why this data is included otherwise.
Response 8: The argument purported by Henriksen et al’s JMBR paper (lines 44 to 48; ref 9-12) provides a putative mechanism of action resulting from the entero-osseous axis following ingestion of a fortified MBPM. However, n.s.d. in treatment effect was observed, reported accordingly and discussed (lines 249 to 260). Lack of effect was somewhat of a surprise to us and in conflict with previous observations and should be reported. With no difference between treatment, i.e. the bioactive components contained in MBPM, this outcome limited further analysis of association.
Point 9: 9.Was there a difference in effect of the test drink in women taking calcium supplements? Numbers may be too small to establish this
Response 9: We thought on similar lines in preparation for this and the further longitudinal study. Extant evidence suggest a differential effect to treatment with variable (principally low) habitual calcium intake by supplementation (e.g. Manson and Bassuk review in JAMA DOI: 10.1001/jama.2017.21012) versus the counterpoint on the effectiveness of such a strategy as reviewed recently by Ian Reed’s group; DOI: 10.1210/jc.2019-00111). So no, the effect of habitual supplemental calcium was not examined and, as you state, would be too small in number to examine robustly.
Of course, the prescribed dietary control did not allow for supplements of any kind to interfere during the experimental phase of the study.
Reviewer 2 Report
This manuscript investigated the acute change in biomarkers of bone turnover following late evening feeding calcium-fortified diet in post-menopausal women with osteopenia. The authors found that the plasma PTH level and the nocturnal peak of bone resorption marker CTX was temporally reduced in the nocturnal feed the calcium-fortified diet group, compared to control diet group. In addition, the uNTX level also significantly reduced in the nocturnal feed the calcium-fortified diet group. Although the participant recruited in the study only 8 for each intervention group, the results are convincing and interesting, looking forward to the author’s 24w study results.
Comments:
1. In the manuscript, the authors indicates SD in the form of XX(yy), suggesting whether to change to the general XX±yy representation.
2. In Figure 5, I do not understand why the same sample number appears in both the CON and MBPM groups. F159, F542 ... is the urine sample number of the experimental individual? Please give a detailed explanation.
3. Two clerical errors appear on line 13 “In CON participants received 804”, not 8041; and line 245, “marker”, not market.
Author Response
Response to Reviewer 2 Comments
Comments and Suggestions for Authors
This manuscript investigated the acute change in biomarkers of bone turnover following late evening feeding calcium-fortified diet in post-menopausal women with osteopenia. The authors found that the plasma PTH level and the nocturnal peak of bone resorption marker CTX was temporally reduced in the nocturnal feed the calcium-fortified diet group, compared to control diet group. In addition, the uNTX level also significantly reduced in the nocturnal feed the calcium-fortified diet group. Although the participant recruited in the study only 8 for each intervention group, the results are convincing and interesting, looking forward to the author’s 24w study results.
Response: Thank you for the positive review. To clarify, the study design is a within-subject repeated measure design of 16 subjects, not 16 subjects divided into 2 groups of 8.
We hope to submit the 24w study within the few weeks.
Comments:
1. In the manuscript, the authors indicates SD in the form of XX(yy), suggesting whether to change to the general XX±yy representation.
2. In Figure 5, I do not understand why the same sample number appears in both the CON and MBPM groups. F159, F542 ... is the urine sample number of the experimental individual? Please give a detailed explanation.
3. Two clerical errors appear on line 13 “In CON participants received 804”, not 8041; and line 245, “marker”, not market.
Point 1: In the manuscript, the authors indicates SD in the form of XX(yy), suggesting whether to change to the general XX±yy representation.
Response 1: We find individual Journals to prefer/prescribe preferred formats. Nutrient appears to allow for both, but are happy to change as requested. Corrected in the revision.
Point 2: In Figure 5, I do not understand why the same sample number appears in both the CON and MBPM groups. F159, F542 ... is the urine sample number of the experimental individual? Please give a detailed explanation.
Response 2: As stated in the text (line 216), the inter-individual response varied widely in this population and appropriate to provide individual data in addition to the overall response. The numbers identify each individual and, thereby the response to the 2 treatment conditions (MBPM and CON) (also see Response above).
To clarify your point for the reader the legend to Figure 5 has been modified in the revision.
Point 3: Two clerical errors appear on line 13 “In CON participants received 804”, not 8041; and line 245, “marker”, not market.
Response 3: Thank you. Corrected in the revision.
We have also corrected all other typo’s and grammatical errors within the revised m/s.